# DEEP CONVOLUTIONAL RECURRENT NEURAL NETWORK FOR SHORT-INTERVAL EEG MOTOR IMAGERY CLASSIFICATION

## ABSTRACT

In this paper, a high-performance short-interval motor imagery classifier is presented that has good potential for use in real-time EEG-based brain-computer interfaces (BCIs). A hybrid deep Convolutional Recurrent Neural Network with Temporal Attention (CRNN-TA) is described that achieves state-of-art performance in four-class classification (73% accuracy, 60% kappa, 3% higher than the winner of the BCI IV 2A competition). An adaptation of the guided grad-CAM method is proposed for decision visualization. A novel EEG data augmentation technique, shuffled-crossover, is introduced that leads to a 3% increase in classification accuracy (relative to a comparable baseline). Classification accuracies for different windows sizes and time intervals are evaluated. An attention mechanism is also proposed that could serve as a feedback loop during data capture for the rejection of bad trials (e.g., those in which participants were inattentive).

## 1 INTRODUCTION

The use of Deep Neural Networks (DNN) for the challenging task of interpreting and classifying the noisy and idiosyncratic signals produced by electroencephalography (EEG) is beginning to show promising results. Some of the best performing models, such as EEGNet (Lawhern et al., 2018) and ShallowFBCSP net (Schirrmeister et al., 2017), have been used in end-to-end solutions that require little or no signal pre-processing.

In this article, we propose an architecture for the classification of Motor Imagery (MI) signals; *viz.*, the EEG signals elicited when participants imagine (rather than physically perform) a motor action, a topic of interest due to its potential as an interface modality, particularly for those with motor impairments.

Traditional approaches to EEG signal classification, such as Common Spatial Pattern (CSP) (Pfurtscheller & Neuper, 2001) and Filter Bank Common Spatial Patterns (FBCSP) (Ang et al., 2008), have the advantage that they can interpret the EEG signals elicited by each MI event in terms of their spatial correlations (e.g., by decomposing eigenvalues by channel (Pfurtscheller & Neuper, 2001)). Unfortunately, although these approaches yield high performance in two-class MI classification (e.g., left vs right), achieving good performance in four-class classification has been elusive, and it therefore seems unlikely that they can underpin more sophisticated BCI systems. Furthermore, CSP-based approaches are not end-to-end solutions, requiring a number of intermediate stages to achieve high performances. For instance, CSP approaches are sensitive to noise (Devlaminck et al., 2011) and require *a priori* feature selection, either automatically or through careful analysis to identify critical features (such as optimal frequency bands). Furthermore, potentially useful information (in terms of increasing classification accuracy) could be lost when the signal is decomposed into covariance matrices and eigenvectors that other approaches may be able to capitalize upon.

Approaches using the Discrete Wavelet Transform (DWT) (Naeem et al., 2006; Yusoff et al., 2018) also typically require pre-processing steps for the interpretation of MI events in the frequency domain to improve signal-to-noise ratio (SNR), such as spatial/temporal filtering and artefact removal. However, combining DWT with other approaches has the potential to enable more sophisticated analyses to be performed, and may help elucidate the association between observed EEG signals and the specific events, such as MI actions, that generated them. Nevertheless, DWT still requires a

careful selection of the mother wavelet. A dimensionality reduction process is also commonly used, or features are calculated as metrics of the decomposition coefficients, implying a loss of potentially useful information and representing an obstacle to the construction of an end-to-end solution. Furthermore, the DWT features that would lead to the highest classification performance are best detected using longer trial windows, representing a challenge for their use in real-time systems.

DNN interpretability is an emerging field that enables researchers to examine network decisions. Techniques to accomplish this include gradient-weighted Class Activation Mapping (grad-CAM) (Selvaraju et al., 2017) and attention mechanisms, used previously in computer vision and natural language processing domains (Vaswani et al., 2017; Bahdanau et al., 2014). At the time of writing, grad-CAM has not been used for DNN-mediated MI classification, but has been used successfully in a small number of EGG signal classification studies using wavelet transformation features (Andreotti et al., 2018). Furthermore, only a handful of studies have employed attention mechanisms for MI classification: for instance, Zhang and colleagues (Zhang et al., 2019) achieved classification accuracy comparable with state-of-art methods using a Convolutional Recurrent Attention Model (CRAM) that uses time domain signals and incorporates an attention mechanism.

Data augmentation is a commonly used technique that aims to improve the accuracy of a trained model and reduce model overfitting by expanding the volume of training data. Most often, new training data are generated in the image domain by cropping, flipping, rotating, adding noise, or modifying color properties of already existing training data (Cireşan et al., 2010; Zeiler & Fergus, 2014). In this article, we develop a similar approach to expand EEG training data and observe increased classification accuracy. We also experimentally determine the minimum time interval and optimal time window needed to confer state-of-the-art classification accuracy.

## 2 METHODS

### 2.1 DATASET

The public BCI Competition IV 2A EEG dataset (Brunner et al., 2008) was used. These data were collected from 9 participants and have four-classes (i.e., four MI movements: left hand, right hand, feet, tongue). Participants completed 250 trials (thus there were 2250 signals captured in all). Each trial was about 9s in duration. At the start of each trial, a fixation cross was presented ($t = 0$). After 2s, an arrow was presented to instruct participant which of the four imagined movements to perform (arrow pointing left for left hand, right for right hand, down for feet, and up for tongue). Participants were instructed to maintain the imagined movement until the arrow disappeared (4s later, at $t = 6$). Next, a random inter-trial break between 2s and 4s occurred, during which a blank screen was shown. Data were collected from 22 silver chloride (Ag/AgCl) electrodes, positioned in the international 10-20 system (Klem et al., 1961). Electrodes were sampled at 250 Hz. Data were band-pass filtered from 0.5 Hz to 100 Hz and notch filtered at 50 Hz to eliminate noise generated from electrical interference. The 4s portion of each signal in which MI was performed was extracted for use here, giving 1000 samples per trial at the sample rate of 250 Hz. From here onwards, when using the term trial, we refer to the 4s period in which the movement was imagined only. The dataset was divided into a training (80) and validation (20) sets. A testing set, provided separately by the BCI competition organizers, was used to evaluate the generalizability of the model since it was unseen during training and validation.

### 2.2 APPARATUS

An NVIDIA GeForce GTX 1050 GPU was used for calculating model parameters (stochastic gradient descent and model updating, see later). CUDA 8 and PyTorch (Paszke et al., 2019) were used for model implementations.

### 2.3 DATA REPRESENTATION

The proposed models were evaluated using the four-class BCI IV 2A Competition dataset (described above). For each participant, raw EEG data were stored in three-dimensional tensor, $X$. The first dimension was trial $t \in [1..T]$ (where T represents the total number of trials per participant, here 250), the second dimension was electrode number $e \in [1..E]$ (where $E$ represents the total number

of electrodes, here 22), and the third dimension was sample $s \in [1..S_{max}]$, where $S_{max}$ represents the number of samples in the longest trial for that participant, up to 4×250. A label vector $y$ contained the true class for each trial, $j \in [1..T]$, wherein each value was a member of set $C = \{0, 1, 2, 3\}$ corresponding to the four classes, left hand (0), right hand (1), feet (2) and tongue (3), thus $|C| = 4$.

## 2.4 Augmentation by shuffling crops and sliding windows

An augmentation technique that appears to be novel in BCI research was used. New trials were generated by taking two non-overlapping intervals of 500 samples (2s) from each 1000 sample (4s) trial for each participant and grafting them together in reverse order. This had the effect of doubling the volume of data available for training, validation, and testing. Furthermore, for the runs using two seconds samples ($S_{max} = 500$), the four seconds samples ($S_{max} = 1000$) were divided into two-two seconds introducing double the number of trials. To simulate real-time signals, overlapped cropped intervals were generated from both the original and augmented 4s data signals. Two window sizes were tested, $\omega \in \{200, 400\}$ samples. One forward jump (stride) of $\sigma = 50$ samples (0.2s) was used. Intervals (sub-vectors) from within each trial for each participant were generated for the sequence of sample indices [r..r+w], where r starts at 1 and was increased by $\sigma$ until no further complete intervals could be collected. These sub-vectors are referred to later as time-slices. For each configuration, $P$ time slices were created (Eq. 1).

$$P = \left\lceil \frac{S_{max} - \omega}{\sigma} \right\rceil \tag{1}$$

## 2.5 Convolutional Recurrent Neural Network with Temporal Attention (CRNN-TA)

### 2.5.1 Convolutional Neural Network (CNN) Blocks

CNN Block 1: First, a standard convolution layer that operates simultaneously on the samples at each time index from all electrodes was used. Its purpose is to find correlations between electrodes. A kernel size of $E \times 1$ was used. After convolution, batch normalization was applied with momentum set to 0.993. A Leaky ReLU activation function (Maas et al., 2013) was then used, a variant of ReLU that can deal with negative values (with gradient constant, $\alpha$, set to 0.01). Next, an average pool with kernel size 1×3 and stride size 1×3 was added to reduce feature dimensionality.

CNN Block 2: Two replicas of the second block type were added to deepen the model, such that each block comprised separable convolutional layers (Chollet, 2017) to reduce the number of trainable parameters. Batch normalization and Leaky ReLU were again added. Following the nonlinear activation function, an average pool layer with kernel size 1×3 and stride 1×3 was used. Finally a dropout layer with dropout probability $p = 0.5$ was added. This generated encoded features (feature maps), which are to be denoted $M$.

The CNN blocks operated as a time-invariant system, wherein different time-slice positions were treated in the same way. In other words, the features in each time slice were stacked on top of each other; e.g., the 4D input tensor (of order $T \times E \times P \times S$) was transformed to a 3D matrix (of order $TP \times E \times S$). The encoded features are then transformed back for the RNN block (see below) to learn time-slices dependencies.

### 2.5.2 Recurrent Neural Network (RNN) GRU Block

Two stacked Gated Recurrent Unit (GRU) layers with 64 hidden units were used after the CNN blocks described above, operating on the time-slices, such that the recurrent operation is applied between each second on the encoded features obtained from the CNN blocks. The 2D latent feature matrix (of order $P \times H$) is defined as $F \in \mathbb{R}^{PH}$ where $P$ is the number of time slices, and $H$ is the number of units in each hidden latent representation. The output of the GRU contains the output for each time slice (rows, $P$) for each of the hidden units (columns, $H$).

### 2.5.3 EXCITATION AND SQUEEZE TEMPORAL ATTENTION

Since participants were asked to maintain each imaginary movement for 4s over 250 trials, their ability to concentrate may have fluctuated from trial-to-trial. Consequently, data in which participants had lost focus will inevitably feature in the dataset. In practice, providing feedback to notify participants whose concentration wanes during training and then labelling (and potentially discarding) data from these trials may improve compliance and enable training data quality to be improved. To reduce the contribution of data corresponding to periods of lapsed concentration, a temporal attention mechanism was used based upon squeeze and excitation (Hu et al., 2018). However, the operation was reversed in relation to the way it is normally used, and is therefore better described as excitation and squeeze. The proposed method requires only around 800 learnable parameters, compared with 17,000 in CRAM (Zhang et al., 2019). The average for each time-slice, $z_i$, was then calculated (Eq. 2).

$$z_i = \frac{1}{H} \sum_{j=1}^{H} f_{i,j}, i = 1..P \tag{2}$$

Next, $z$ (of order $P \times 1$) was excited using a nonlinear ReLU activation function applied (Eq. 3), where $W_1$ is a matrix of order $P \times 64$ containing learnable weights (i.e., mapping to 64 neurons).

$$z' = ReLU(zW_1) \tag{3}$$

The vector produced,$z'$, is of order 1×64. To acquire an attention vector for each time-slice, and learn nonlinear dependencies between time-slices $z'$ is squeezed using a sigmoid activation function (Eq. 4) where $W_2$ is a weight matrix of order $64 \times P$.

$$a = sigmoid(z'W_2) \tag{4}$$

The attention vector, a (of order $1 \times P$), is multiplied element-wise by each of the $H$ columns of $F$ (where each column is of order $P \times 1$) to emphasize the contribution of the features in of each time-slice to the model decision, rescaling the hidden features to create $\hat{F}$ (Eq. 5, where $\circ$ denotes Hadamard product).

$$\hat{F}_{1..p,j} = a' \circ F_{1..p,j}, j = 1..H \tag{5}$$

### 2.5.4 FULLY CONNECTED LAYER

The output of the attention mechanism, Eq. 5, was then flattened to form an input vector that is connected to $C$ output neurons, where $|C|$ is the number of classes. A $softmax$ activation function (Eq. 6) was then applied to the output neurons for classification, where $W_3$ is a weight matrix of order $64P \times |C|$.

$$\hat{y} = softmax(W_3\hat{F}) \tag{6}$$

### 2.6 TRAINING, TESTING AND VALIDATION

A cross-entropy function was used to calculate the loss of the model. During training, the model attempts to minimize loss and the model was penalized when the probability of the predicted class differed from the true class. Binary cross-entropy is defined as shown in Eq. 7.

$$-(y \log_2(p) + (1 - y) \log_2(1 - p)) \tag{7}$$

Where $y$ is the true class (binary 0 or 1) and $p$ is the probability associated with the class prediction. Where multi-class classification is required, loss $L$ is calculated for each class $i$, where each is a value $c \in C$, separately and losses are summed to obtain a single loss value (Eq. 8) for each observation.

$$L = -\sum_{i=1}^{|C|} y_i \log_2(\hat{y}_i) \tag{8}$$

The weights and biases were updated using batch gradient descent. Batch size was set to 20, thus weights were updated 20 times per epoch (see below), with the total loss being the average loss over 20 trials. The final predicted class label, $\hat{y}_i$, is calculated as the class with the maximum probability. Adam (Kingma & Ba, 2014), was used for weight optimization and updating, with weight decay set to 0.4 to introduce L2 regularization (Krogh & Hertz, 1992). A total of 800 epochs were used for training. The reported accuracies are the mean of five runs, similar to five-fold cross validation in which the training samples are non-overlapped and randomly selected for each run. Accuracies were calculated using the unseen testing set.

## 2.7 MODIFIED GUIDED GRAD-CAM

Selvaraju and colleagues (Selvaraju et al., 2017) presents a generalization of Class Activation Mapping (CAM) (Zhou et al., 2016) that is usable for any CNN-based architecture. These visualization techniques were initially developed to identify image regions that contribute to model decisions. However, whilst image features may be readily interpretable, where EEG signals are used, visual inspection will not be meaningful. Consequently, the approach used here has two components. First, signal intervals that contribute significantly to the classification of a specific class are approximately localized, enabling them to be used in a feedback loop when the model is being trained. Secondly, identifying the spatial correlation (the relationship) between the electrodes may be exploitable to minimize the number of electrodes required in real life (e.g., to simplify BCI hardware). To accomplish this, some modifications to the standard method are required.

In the general definition of CAM, the last convolutional layer before the $softmax$ in any CNN based architecture generates $K$ feature maps, $M^k \in \mathbb{R}^{uv}$ where $u$ (rows) and $v$ (columns) represent the activated units in a 2D plane indexed by $i$ and $j$, e.g. $M_{ij}^k$ is the activated unit at location $(i,j)$ of the $k^{th}$ feature map. Global Average Pooling (GAP) is applied to $M^k$, followed by a linear transformation to obtain a weighed score $Y^c$ for each class $c$, and $W_4$ is a weight matrix of order $K \times |C|$. For ease of reading we will refer to each weight connecting a $k^{th}$ feature map with a $c^{th}$ class as $w_k^c$ (Eq. 9).

$$Y^C = \sum_{k=1}^{K} w_k^c \frac{1}{Z} \sum_{i=1}^{u} \sum_{j=1}^{v} M_{i,j}^k \tag{9}$$

In the proposed model, the spatial correlation between electrodes is encoded in the first convolutional layer, resulting in a 1D matrix with $v$ samples for each feature map $k$ (collapsed dimension after convolving over electrodes). In order to preserve the electrode information and apply Grad-CAM to the EEG signals per electrode (1D matrix), instead of using a GAP, an average pool was applied on $M^k$, so the CAM can now be represented (Eq. 10).

$$Y^C = \sum_{k=1}^{K} w_k^c \frac{1}{Z} \sum_{i=1}^{v} M_i^k \tag{10}$$

To make the CAM weighted by gradients, instead of the final layer learned weights [19], the gradient of $Y^c$ is calculated with respect to the feature maps $M_i^k$ (Eq. 11).

$$\alpha_k^c = \frac{1}{Z} \sum_{i=1}^{v} \frac{\partial Y^c}{\partial M_i^k} \tag{11}$$

Where the partial linearization $\alpha_k^c$ is obtained by average pooling the gradients acquired through back-propagating from the feature maps of the target class $c$. Finally, the heatmaps generated are defined as the combination of the feature maps and are also followed by a $ReLU$ to acquire only the features that contribute positively (i.e., more important electrodes), Eq. 12:

Table 1: Independent participant's classification accuracy in four different configurations (windows size, stride, number of samples). Baseline column without excitation and squeeze or augmentation. Proposed method column using excitation and squeeze with data augmentation. Reported accuracies on unseen test data from BCI IV 2A. The four seconds samples $S_{max}$ were divided into two-two seconds for the proposed method.

| $\omega$ | $\sigma$ | #Samples | Baseline Mean Acc (SD) | CRNN-TA Mean Accuracy SD |
|---|---|---|---|---|
| 400 | 50 | 1000 | 0.68 (0.01) | |
| 200 | 50 | 1000 | 0.65 (0.01) | |
| 400 | 50 | 500 | 0.65 (0.01) | **0.70** (0.12) |
| 200 | 50 | 500 | 0.65 (0.01) | **0.73** (0.08) |

Table 2: A comparison between the best performing method from BCI Competition IV 2A, measured in Cohen's Kappa ($\kappa$), CRAM, and the proposed model (last column, labelled CRNN-TA). For each of the 9 participants, the best preforming model is highlighted in bold.

| Participant | 1st Place | 2nd Place | 3rd Place | CRAM | CRNN-TA |
|---|---|---|---|---|---|
| 1 | 0.68 | 0.69 | 0.38 | 0.69 | **0.70** |
| 2 | **0.42** | 0.34 | 0.18 | 0.29 | 0.34 |
| 3 | 0.75 | 0.71 | 0.48 | 0.68 | **0.79** |
| 4 | 0.48 | 0.44 | 0.33 | 0.34 | **0.51** |
| 5 | 0.40 | 0.16 | 0.07 | 0.09 | **0.45** |
| 6 | 0.27 | 0.21 | 0.14 | 0.30 | **0.39** |
| 7 | 0.77 | 0.66 | 0.29 | 0.57 | **0.80** |
| 8 | **0.75** | 0.73 | 0.49 | 0.49 | 0.72 |
| 9 | 0.61 | 0.69 | 0.44 | 0.56 | **0.66** |
| **Mean** | 0.57 | 0.52 | 0.31 | 0.45 | **0.61** |

$$L^c_{(grad-CAM)} = ReLU \left( \sum_{k=1}^{k} a_k^c M^k \right) \tag{12}$$

The heatmap generated for the first convolutional layer, $L^c_{(grad-CAM)}$, will have the same size as the feature map (i.e., 22×400 in the suggested model). After applying the convolution operation at the spatial layer (with kernel size 22×1), the feature maps become 1D (1×400).

## 3 RESULTS

Results for each configuration are shown in Table 1. For the baseline, no statistically significant difference in mean accuracy between the four configurations was found: $F(3, 241) = 0.92$, $p > 0.01$. However, the configuration $\omega = 400$, $\sigma$ = 50%, $S = 1000$ yielded 68% average accuracy, 3% higher than the remaining groups. Between the suggested model and the baseline for the bottom two configurations, no statistically significant difference in mean accuracy between the four configurations was found: $F(3, 32) = 0.50$, $p > 0.01$. The final configuration ($\omega = 200, \sigma = 50, S = 500$) yielded the highest accuracy of 73%, an improvement of +5% compared to the best baseline score +8% over the same configuration in the baseline.

The suggested model has the highest performance as shown in Table 2 where it outperforms the 1st place winner by 4% and CRAM by 6%.

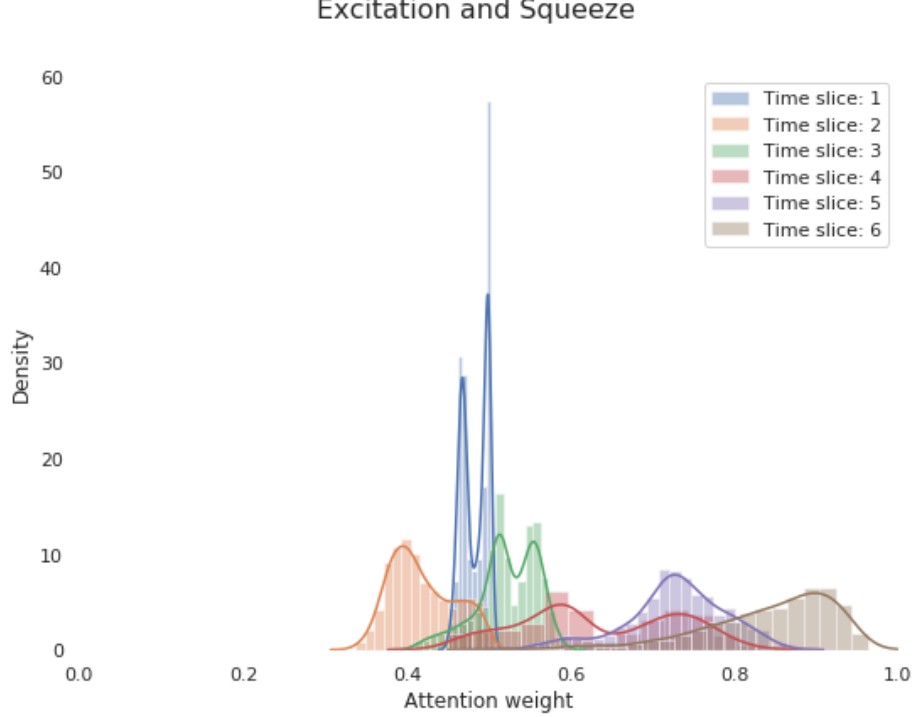

Figure 1: Excitation and squeeze density plot showing weight distribution across all trials for participant 3 with number of time slices $P = 6$ where attention weights (x-axis) were calculated using Eq. 5 and y-axis represents the density of the distribution

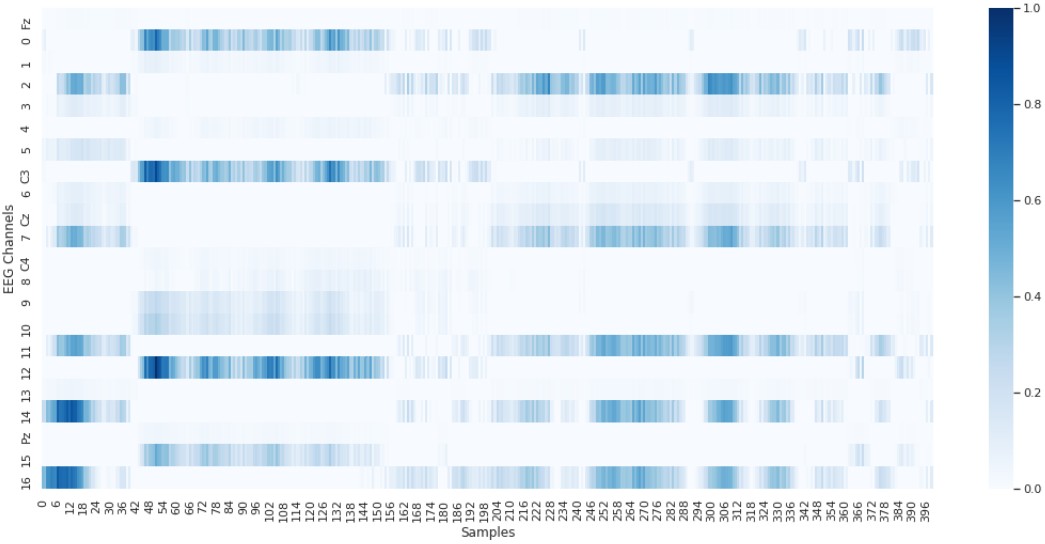

Figure 2: Example heatmap produced by the guided grad-CAM looking at the first convolutional layer, corresponding to participant 3 imagining a right hand movement ($\omega = 400$ and $S = 500$); note that the features were encoded in a time invariant manner as discussed earlier

## 4 DISCUSSION

In this study, two different window sizes ($\omega = 200$ and $\omega = 400$) using either $S = 500$ or $S = 1000$ (2s or 4s intervals) were supplied to the model for training and testing, and results reported are for the unseen test data. The model achieved mean $\kappa = 0.60 \pm 0.02$, representing performance that exceeds state-of-art CRAM ($\kappa = 0.45$), and the winner of BCI IV 2A competition ($\kappa = 0.57$), as summarized in Table 2. The baseline yielded identical performance using $\omega = 200$ or $\omega = 400$ with $\sigma = 50$ and $S = 500$ (Table 1: Baseline), implying that classification could be performed twice as quickly since it used half as much data (i.e., window size 400 over a 4s interval would require a delay of 4s and an input of 1.6s input per one step of classification, compared with a 2s delay and 0.8s input per one step of classification), representing slightly better than the current real-time EEG based classification applications. A further advantage of shorter windows and intervals is that less memory is required to store the signals, further increasing the potential for its use in real-time or embedded BCIs. The data augmentation method proposed, in which extra training samples were generated by exchanging the first and last 2s intervals (note: this was not done to the final testing set used to evaluate the model's performance) led to the highest classification accuracy, with a mean of 70% for the configuration $\omega = 400$, $\sigma = 50$, $S = 500$ and 73% for configuration $\omega = 200$, $\sigma = 50$, $S = 500$, suggesting that further expanding this augmentation technique could potentially further increase performance without requiring that more data is collected/used for training. The attention mechanism proposed (excitation and squeeze, Fig. 1) provided an informative time slice interpretation that highlights the contribution of each slice to the model's decision. The attention weights observed seem to indicate greater attention density for the latter three time slices (4..6). This approach is therefore potentially useful in architectures with a recurrent layer, since it is (relatively) computationally inexpensive, with only 800 learned parameters learned, and the attention vectors produced are potentially useful in a feedback loop. Guided grad-CAM heatmaps (Fig. 2) show a higher contribution of selected channels between the four classes, and highlight which samples contribute more significantly to the model's decision. The guided grad-CAM approach proposed seems to offer positive preliminary results but further investigation is needed to validate the outcome of the grad-CAM (e.g., by removing the channels that contributed least) since it is not possible to verify visually.

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
