# OpenReview forum: "Deep convolutional recurrent neural network for short-interval EEG motor imagery classification"
_ICLR.cc/2022/Conference — ICLR 2022 Submitted_

### Official Review · Reviewer_uzaX · 2021-11-01

**Correctness:** 2
**Technical Novelty And Significance:** 2
**Empirical Novelty And Significance:** 2
**Recommendation:** 3
**Confidence:** 2

**Main Review:**

Pros.

-An interesting data augmentation algorithm for EEG time-series classification.
-A class activation mapping approach for EEG data is presented.

Cons. and comments

-The paper presentation is poor. For example, the introduction is confusing, lacks suitable state-of-the-art analysis, and does not provide the proposal's paragraph.
-The results should include other motor imagery databases. The influence of the number of EEG channels and the inter and intrasubject variability must be studied to test the availability of a "short interval." For example, see:
https://academic.oup.com/gigascience/article/6/7/gix034/3796323
https://physionet.org/content/eegmmidb/1.0.0/
-The mathematical details should be enhanced for an iclr paper.
-Please provide the codes and the experimental details.

**Summary Of The Paper:**

The authors introduce a deep learning approach for short-time motor imagery classification using EEG data.  Conventional CNN and RNN - GRU layers are used. Remarkably, a data augmentation strategy and a class-activation mapping approach are presented. Overall, the idea is interesting, but the paper presentation, the mathematical foundation, and the experiments provided are poor. Therefore, more details about the model and other experiments should be carried out to validate the proposal. Besides, the authors claim that short time interval EEG classification is achieved; however, 0.8s windows size does not seem to be a "short interval" compared to other state-of-the-art methods.

**Summary Of The Review:**

A good idea is presented concerning the data augmentation and the cam-based extensions for EEG data. Nonetheless, the short interval claim seems to be ambiguous. Several details regarding the mathematical background and more experiments should be conducted to test the proposal. In addition, the paper presentation needs to be enhanced.

---

### Official Review · Reviewer_tCsE · 2021-11-02

**Correctness:** 2
**Technical Novelty And Significance:** 1
**Empirical Novelty And Significance:** 1
**Recommendation:** 1
**Confidence:** 5

**Main Review:**

It is the opinion of the reviewer that following are the strengths and weaknesses of the paper.

Strengths:
- The paper is generally well-written and structured.
- Good results are achieved.

Weaknesses:
- The proposed method is quite simple (Conv RNN + att), unfortunately with no novelty or innovation. Similar methods have been widely used in the past, even in the field of EEG representation learning, e.g. Zhang et al., Classification of Hand Movements from EEG using a Deep Attention-based LSTM Network, 2020
- Only one dataset (BCI IV 2A) is used. Therefore, it is very unclear how the method generalizes to other datasets.
- No in-depth analysis of the length of the window sizes (i.e., "short interval") is carried out.
- The model details are not all give, and so the work is not reproducible.
- A comprehensive comparison with other methods in the field is missing.

Based on the shortcomings mentioned above, unfortunately the paper is very far from the level of ICLR.

**Summary Of The Paper:**

The paper proposes a short-interval MI classification system. The model is a convolutional RNN with temporal attention.

**Summary Of The Review:**

-

---

### Official Review · Reviewer_M3gg · 2021-11-05

**Correctness:** 3
**Technical Novelty And Significance:** 2
**Empirical Novelty And Significance:** Not applicable
**Recommendation:** 1
**Confidence:** 5

**Main Review:**

Strengths:
1. The topic of EEG classification is meaningful.

2. The modified class activation mapping (Grad-CAM) is interesting to me.

Weaknesses:
1. The representation is not clear. To me, the challenges, motivation, solutions, or contributions are not clearly expressed. For example,

- Since the title mentioned 'short-interval EEG', then what's long-interval EEG? What's the difference and how do existing studies treat them differently?

- Sec. 1 uses two long paragraphs to show the limitations of traditional approaches, which are all well known. Please condense them into one paragraph.

2. Lack of novelty. Using standard deep learning models (and attention motivation) for EEG recognition is ok in 2018, but now, we expect more technical novelty. In specific, the CNN, RNN, excitation and squeeze, and FUlly-connected layers are very commonly used and intensively studied components in the last 5 years.

3. Weak experiments.
- This work only compared with one baseline, which is not enough. I am also confused that what is exact the baseline mentioned in Table 1? I cannot find descriptions about the baseline's model/structure/citation.

- This work claims its effectiveness by 'outperforming the 1st place winner by 4%' (in Abstract and Results). However, the BCI IV competition is in 2008! It is not amazing that the proposed model is better than a winner in 2008.

- The work lacks lots of studies on DL-based EEG analysis in recent 3 years. Please find my recent works and compare them.

**Summary Of The Paper:**

This paper proposes a model combining convolutional operation and recurrent operation along with temporal attention for EEG classification.

**Summary Of The Review:**

As I mentioned in the above weakness, this manuscript is not presented well, the technical novelty is not significant, and the experiments are not extensive. I suggest a strong rejection.

---

### Decision · Program_Chairs · 2022-01-20

**Decision:**

Reject

**Comment:**

This paper develops a deep convolutional network with RNN layers and
a new data augmentation method for EEG motor imagery classification.

Reviewers agreed that the paper was not very clearly written, and that
without comparisons to other related methods or at least demonstration
of the importance of each of the components of the model (through for
example ablation analyses), it was hard to understand the generality
of the approach.

The authors did not respond to the reviews, so I am recommending not
accepting this paper.